# Budget impact analysis of HARMONIC FOCUS™+ Shears for mastectomy and breast-conserving surgery with axillary lymph node dissection compared with monopolar electrocautery from an Italian hospital perspective

**Alessandra Piemontese**[1]*, **Thibaut Galvain**[2], **Lirazel Swindells**[3], **Vito Parago**[1], **Giovanni Tommaselli**[4], **Nadine Jamous**[1]

1 EMEA Health Economics & Market Access, Johnson & Johnson Medical Devices Companies, Diegem, Belgium, 2 Global Health Economics & Market Access, Johnson and Johnson Medical Devices, New Brunswick, NJ, United States of America, 3 Costello Medical Consulting Limited, Cambridge, United Kingdom, 4 Global Medical Affairs, Johnson & Johnson Medical Devices Companies, Cincinnati, OH, United States of America

* APiemont@its.jnj.com

## Abstract

### Background

Mastectomy or breast conserving surgery, both with axillary lymph node dissection, are common treatments for early-stage breast cancer. Monopolar electrocautery is typically used for both procedures, despite evidence of improved clinical outcomes with HARMONIC FOCUS™+. This analysis evaluated the budget impact of adopting HARMONIC FOCUS™+ versus monopolar electrocautery for patients undergoing these procedures from an Italian hospital perspective.

### Methods

Total costs for an annual caseload of 100 patients undergoing mastectomy or breast con-serving surgery, with axillary lymph node dissection, with either the intervention or compara-tor were calculated. Italian clinical and cost input data were utilised. The analysis included costs for the device, operating room time, postoperative length of stay, treating seroma and managing postoperative chest wall drainage. Deterministic and probabilistic sensitivity anal-yses assessed uncertainty of model input values. Two scenario analyses investigated the impact of conservative estimates of postoperative length of stay reduction and daily hospital cost on the simulated cost difference.

### Results

HARMONIC FOCUS™+ achieves annual savings of EUR 100,043 compared with monopo-lar electrocautery, derived from lower costs for operating room time, postoperative length of

**Data Availability Statement:** All relevant data are within the manuscript and its Supporting Information files.

**Funding:** This study was funded by Johnson & Johnson. Alessandra Piemontese, Thibaut Galvain, Vito Parago, Giovanni A. Tommaselli and Nadine Jamous are all employees of Johnson & Johnson. Support for third-party writing assistance for this article, provided by Costello Medical, UK, was funded by Johnson & Johnson in accordance with Good Publication Practice (GPP3) guidelines (http://www.ismpp.org/gpp3).

**Competing interests:** I have read the journal's policy and the authors of this manuscript have the following competing interests: TG, NJ, VP, AP and GT are paid employees of Johnson & Johnson. At the time of writing the manuscript, LS was a paid employee of Costello Medical, UK, who was contracted by Johnson & Johnson to conduct and assist in developing the model and provide medical writing support in preparation of the manuscript. This does not alter our adherence to PLOS ONE policies on sharing data and materials.

stay and seroma and postoperative chest wall drainage management, offsetting the incremental device cost increase (EUR 43,268). Cost savings are maintained in scenario analyses and across all variations in parameters in deterministic sensitivity analysis, with postoperative hospital stay costs being key drivers of budget impact. The mean (interquartile range) cost savings with HARMONIC FOCUS™+ versus monopolar electrocautery in probabilistic sensitivity analysis are EUR 101,637 (EUR 64,390–137,093) with a 98% probability of being cost saving.

## Conclusions

The intervention demonstrates robust cost savings compared with monopolar electrocautery for mastectomy or breast conserving surgery, with axillary lymph node dissection, in an Italian hospital setting, and improved clinical and resource outcomes. These findings, with other clinical and cost analyses, support HARMONIC FOCUS™+ use in this setting.

## Introduction

Female breast cancer is the most common type of cancer. In 2020, 2.3 million new cases were diagnosed worldwide, accounting for 11.7% of cancer diagnoses among men and women [1]. The incidence of breast cancer across Europe is almost double the global rate [1, 2]. Within Italy, breast cancer comprises 28% of cancer diagnoses in women [3]. High incidence and intensive management strategies for this disease pose a substantial economic burden for hospitals [4, 5]. An Italian registry study found direct medical costs incurred six months prior to diagnosis through 24 months after diagnosis/surgery to be EUR 10,970 per patient, of which 80% were attributed to treatment [4]. Therefore, minimising costs associated with treatment is key in reducing the economic burden of breast cancer.

The 2019 European Society for Medical Oncology (ESMO) guidelines recommend breast conserving surgery (BCS) or mastectomy for the treatment of early-stage breast cancer, according to patient preference and/or characteristics, as well as a sentinel lymph node biopsy (SLNB) to direct intraoperative patient management [6]. Axillary lymph node dissection (ALND) is also further recommended by ESMO guidelines in patients with (micro)metastatic spread [6, 7]. ALND may improve locoregional control and survival because axillary lymph nodes are a common site of metastasis and cancer in these regions is a key indicator of poor prognosis [8, 9]. However, when compared with SLNB alone, ALND has been associated with increased risk of complications including lymphedema and seroma [8, 10], the latter of which affects up to 85% of patients undergoing ALND and is a source of substantial morbidity [11, 12]. Seroma formation may also delay adjuvant treatments such as chemotherapy and, as a result, affect oncological outcomes [12]. Therefore, control of surgical complications in patients who undergo ALND is particularly important.

Monopolar electrocautery, alongside other conventional techniques, is the current standard of care for dissection and haemostasis in breast surgery [13]. However, conventional techniques are associated with limitations, including the risk of incomplete sealing, thermal injury, and complications such as seroma, bleeding and tissue necrosis [13–15]. Using conventional techniques also increases the need to alternate between different instruments in order to achieve effective sealing of different vessel sizes and to minimise surgical smoke production

[14]. Such challenges may extend operating room (OR) times, require costly treatment and slow patient recovery, thereby extending postoperative length of stay (LOS) [16–19].

HARMONIC FOCUS™+ uses ultrasonic energy to simultaneously cut and coagulate tissue and vessels at lower temperatures than conventional techniques. Given these properties, there is potential for its use to reduce OR time through faster haemostasis, less visual obstruction from smoke and fewer instrument changes. These factors may in turn hasten recovery and reduce postoperative LOS [14]. Multiple randomised controlled trials (RCTs) have assessed the clinical value of ultrasonic energy versus conventional techniques for breast procedures and have been included within several meta-analyses [13, 14, 20, 21]. While some economic analyses in favour of ultrasonic energy have been published [22–24], none have been conducted in an Italian setting where the burden of breast cancer is prominent [1].

The objective of this budget impact analysis (BIA) was to evaluate the budget impact of adopting HARMONIC FOCUS™+ versus monopolar electrocautery for patients undergoing mastectomy with ALND or BCS with ALND from an Italian hospital perspective.

## Materials and methods

A BIA was designed to estimate the cost impact of using HARMONIC FOCUS™+, compared with monopolar electrocautery for mastectomy with ALND and BCS with ALND procedures, for a hypothetical annual caseload of patients from an Italian hospital perspective, using data from the published literature. The BIA was conducted in line with ISPOR BIA Principles of Good Practice [25], and reported in accordance with the Consolidated Health Economic Evaluation Reporting Standards (CHEERS) publication guidelines [26].

### Population

The study population comprised an assumed annual caseload of 100 patients undergoing mastectomy with ALND (83.3%) or BCS with ALND (16.7%) in an Italian hospital setting, wherein the distribution of procedures was informed by an Italian RCT estimated to be representative of standard clinical practice [16].

### Intervention and comparator

The intervention considered in the BIA for tissue cutting and haemostasis during procedures was HARMONIC FOCUS™+ and the comparator was monopolar electrocautery.

### Model design and structure

The BIA design was informed by an Italian RCT conducted by Iovino et al. in 2012, which compared HARMONIC FOCUS™+ (reported in the publication as 'Harmonic scalpel') with monopolar electrocautery in patients who underwent BCS with ALND or mastectomy with ALND [16]. This source of input parameters is the most recent literature applicable to the local setting and is corroborated by several meta-analyses of studies in the breast cancer population, spanning multiple country settings [13, 20, 21, 27]. All outcomes that were analysed in Iovino et al. 2012 and that could be assigned a unit cost value were considered as model parameters. This comprised OR time, postoperative LOS, duration of chest wall drainage and incidence of seroma [16].

The BIA simulated the total incremental costs associated with a full conversion from the comparator to the intervention over a one-year time horizon, including an analysis of the overall cost difference per patient and for each model parameter (i.e., costs of the device, OR time, hospital stay, seroma treatment and postoperative chest wall drainage management). Between-

treatment differences in total hospital days and seroma episodes for the caseload over one year were also presented. Only costs incurred between the procedure and hospital discharge were considered in the BIA. Since the current Italian hospital remunerative system is based on diagnosis-related group (DRG) tariffs, hospitals are reimbursed through fee for service, making the total inpatient cost relevant for the hospital perspective. Discounting was not applied, given the one-year time horizon of the BIA.

## Model inputs

Targeted literature searches of MEDLINE via PubMed were conducted to identify the most recent published sources for clinical and economic inputs specific to the Italian hospital setting. Iovino et al. 2012 informed all clinical inputs [16], whilst most cost input data were directly obtained from several studies conducted in Italian surgical departments [16, 28, 29]. However, since no local cost data were identified for the treatment of seroma, a micro-costing approach was taken, assuming one aspiration per seroma and 15 minutes of nurse time per aspiration, at a unit labour cost of EUR 26.30 per hour [30, 31]. This approach was validated by author's (G.A.T.) clinical experience. All cost input data were inflated to 2020 EUR values according to GDP deflator values for Italy [32]. All clinical and cost input data used in the primary analysis are reported in Table 1 [16, 28–31].

## Sensitivity and scenario analyses

The robustness of the primary analysis was assessed by deterministic sensitivity analysis (DSA), probabilistic sensitivity analysis (PSA) and scenario analyses. In the DSA, all model input data were varied according to 95% confidence intervals (CIs), or by applying a ±25%

**Table 1. Model inputs for the primary analysis.**

| Model parameter | Model input value | | Source |
|---|---|---|---|
| | HARMONIC FOCUS™+ | Monopolar electrocautery | |
| **Caseload parameters** | | | |
| Number of procedures carried out annual, n | 100 | | Assumption |
| **Clinical parameters** | | | |
| Median OR time, mins | 115 | 120 | Iovino et al. 2012 [16] |
| Median LOS, days | 3 | 5 | |
| Median duration of postoperative chest wall drainage, days | 3 | 5 | |
| Median proportion of patients experiencing seroma, % | 3.3 | 30.0 | |
| **Unit cost parameters[a]** | | | |
| Cost per device, EUR | 551.67 | 118.99 | Iovino et al. 2012 [16] |
| OR time cost per minute, EUR | 7.0 | | Corsi et al. 2016 [28] |
| Hospital stay cost per day, EUR | 655 | | |
| Incremental cost of postoperative chest wall drainage per day, EUR | 42.77 | | Vertuani et al. 2015 [29] |
| Incremental cost of treating seroma,[b] EUR | 7.11 | | Jain et al. 2004 [30], Severi et al. 2012 [31] |

[a]Costs inflated to 2020 EUR values according to the Q4 2020 GDP Deflator values for Italy (104.454) and the following GDP Deflator values: Q1 2012 (96.564) for costs derived from Iovino et al. 2012 and Severi et al. 2012, Q1 2013 (97.693) for costs derived from Vertuani et al. 2015 which reports 2013 data, and Q1 2016 (101.064) for costs derived from Corsi et al. 2016 [32]

[b]micro-costed based on UK data indicating a nurse carries out one aspiration per seroma [30], Italian nurse labour time (EUR 26.30 per hour) [31], and an assumption that 15 minutes of nurse time is required per aspiration (costs of materials used during seroma aspiration were assumed negligible); the approach was validated by clinical expert opinion.

**Abbreviations:** EUR: Euros; GDP: gross domestic product; LOS: length of stay; OR: operating room; UK: United Kingdom.

range of variation for input values where 95% CIs were not available. The PSA was performed using the Monte Carlo method, wherein parameters were simultaneously varied by randomly sampling values for each parameter according to their statistical distribution for 1,000 iterations. The upper and lower bounds and statistical distributions used for each model parameter in the DSA and PSA are reported in S1 Table.

Scenario analyses were conducted to consider the impact of using, within the BIA, alternative data sources available in the literature for 1) the expected reduction in postoperative LOS with the intervention and 2) daily cost of hospital stay. In scenario one, a more conservative reduction of 1.38 days was assumed for the intervention in lieu of the base case estimate of a 2-day reduction. This scenario was informed by a meta-analysis which, although less representative of the target population because reporting on LOS spans across different countries [13], provides a conservative estimate of postoperative LOS reduction for an Italian setting. For scenario two, a conservative daily hospital cost of EUR 416 was applied as reported in Berto et al. 2012 (inflated to 2020 EUR values) for patients who underwent colorectal surgery in Italy [33]. Although not specific to breast surgery, this scenario provides a reasonable, conservative estimate for daily LOS costs in the Italian population (EUR 246 lower than base case assumption), as daily hospital costs are not expected to vary substantially across surgical units.

## Results

### Primary analysis

The results of the primary analysis are presented in Table 2. Total costs for the annual caseload of patients undergoing BCS with ALND or mastectomy with ALND are EUR 345,424 with HARMONIC FOCUS™+ and EUR 445,467 with monopolar electrocautery, yielding potential annual net cost savings of EUR 100,043 (approximately EUR 1,000 per patient). The incremental cost of adopting the intervention in place of the comparator for the caseload (EUR 43,268) is offset by cost savings arising primarily from shorter LOS, but also from reduced costs associated with OR time, treatment of seroma and management of postoperative chest wall drainage. The reduction in LOS and the incidence of seroma with the intervention also results in 200 hospital bed days freed and 26.7 fewer seroma episodes over a year.

**Table 2. Results of the primary analysis.**

| Cost component | Per annual caseload (EUR)[a] | | | Per procedure (EUR) | | |
|---|---|---|---|---|---|---|
| | HARMONIC FOCUS™+ | Monopolar electrocautery | Cost difference[b] | HARMONIC FOCUS™+ | Monopolar electrocautery | Cost difference[b] |
| Device costs | 55,167 | 11,899 | 43,268 | 552 | 119 | -433 |
| OR time costs | 80,823 | 84,337 | -3,514 | 808 | 843 | -35 |
| Hospital stay costs | 196,580 | 327,633 | -131,053 | 1,966 | 3,276 | -1,311[c] |
| Complication costs[d] | 12,854 | 21,598 | -8,744 | 129 | 216 | -87 |
| **Overall costs** | **345,424** | **445,467** | **-100,043** | **3,455** | **4,455** | **-1,000** |

[a]Assumes an annual caseload of 100 procedures (mastectomy with ALND or BCS with ALND)

[b]cost difference calculated as the component value of the intervention minus component value for comparator such that negative values indicate cost saving with the intervention

[c]discrepancy with the difference between intervention and comparator per procedure costs is due to rounding

[d]comprises costs for management and treatment of seroma and postoperative chest wall drainage.

**Abbreviations:** ALND: axillary lymph node dissection; BCS: breast-conserving surgery; EUR: Euros; OR: operating room.

## Sensitivity analyses

The tornado diagram in Fig 1 illustrates the results of the DSA and shows that HARMONIC FOCUS™+ remains cost saving compared with monopolar electrocautery across all variations in parameters. Variation in postoperative LOS has the greatest effect on budget impact,

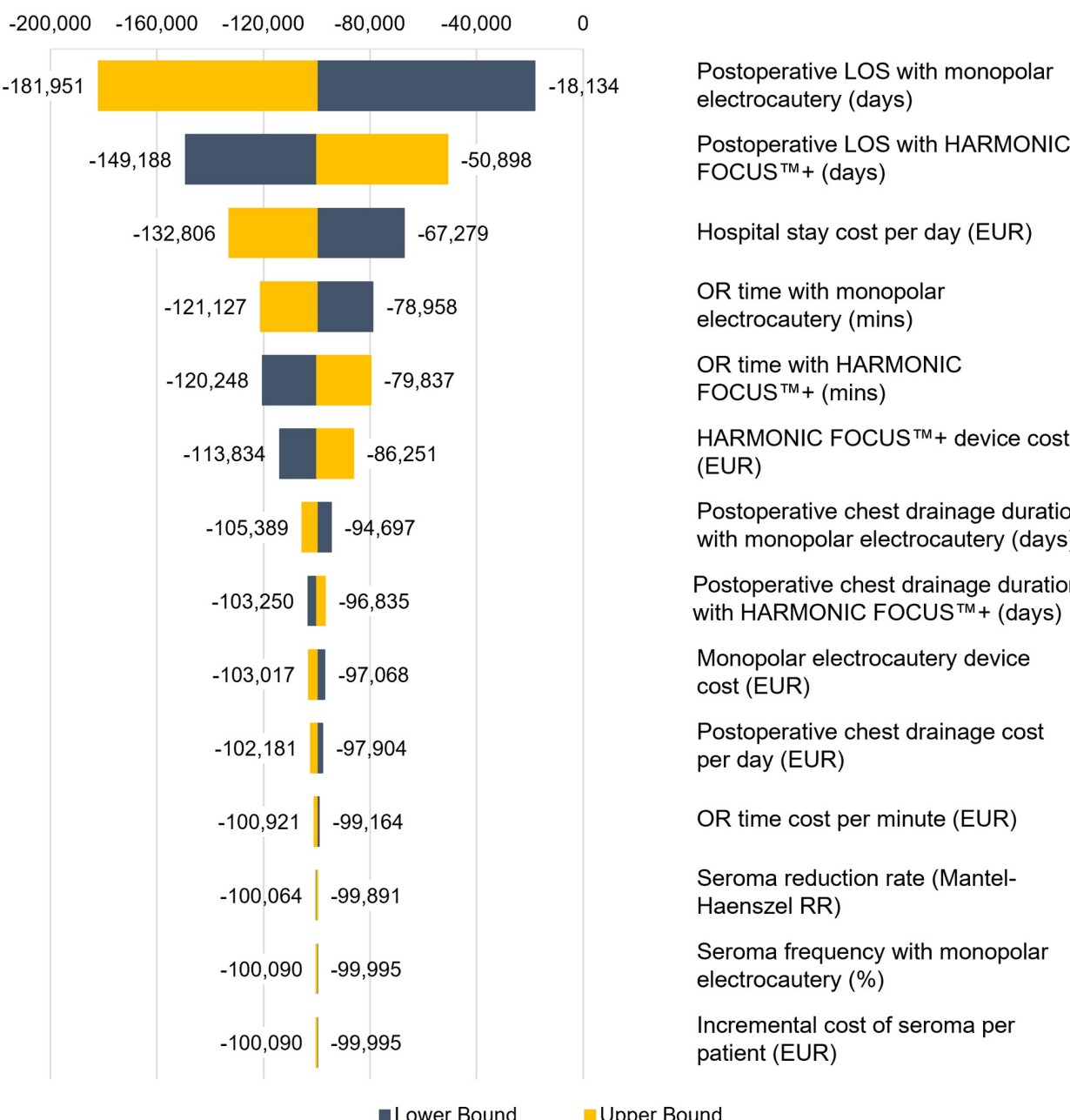

**Fig 1. Tornado diagram of DSA results of the cost impact for the annual caseload.** Negative values indicate cost saving with the intervention compared with the comparator. **Abbreviations:** DSA: deterministic sensitivity analysis; EUR: Euros; LOS: length of stay; OR: operating room; RR: risk ratio.

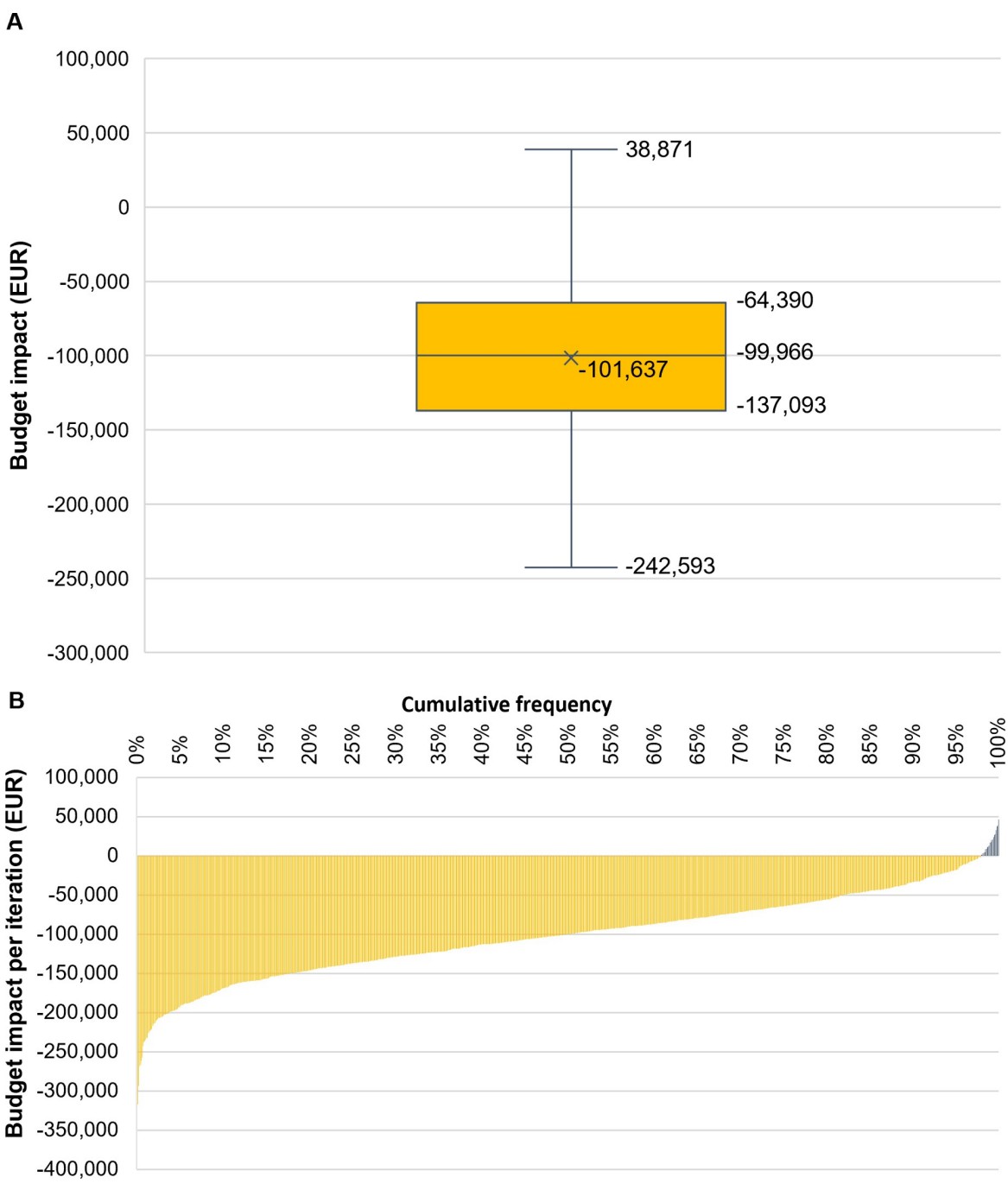

**Fig 2. PSA results of the cost impact for the annual caseload.** Annual caseload PSA results as A) a box-and-whisker plot and B) cumulative distribution graph. Negative values indicate cost saving with the intervention compared with the comparator. **Abbreviations:** EUR: Euros; PSA: probabilistic sensitivity analysis.

whereas factors associated with seroma and postoperative chest wall drainage have a minimal effect. The mean (interquartile range) potential net cost savings with the intervention determined by the PSA for the caseload are EUR 101,637 (EUR 64,390–137,093) and there is a 98% probability of the intervention resulting in cost savings (Fig 2).

**Table 3. Results of the scenario analyses.**

| Analysis | Per annual caseload (EUR)[a] | | |
|---|---|---|---|
| | HARMONIC FOCUS™+ | Monopolar electrocautery | Cost difference[b] |
| Primary | 345,424 | 445,467 | **-100,043** |
| Scenario one[c] | 386,051 | 445,467 | **-59,416** |
| Scenario two[d] | 273,781 | 326,062 | **-52,281** |

[a]Assumes an annual caseload of 100 procedures (mastectomy with ALND or BCS with ALND)

[b]cost difference calculated as the overall cost of the intervention minus the overall cost of the comparator such that negative values indicate cost saving with the intervention

[c]deviates from the primary analysis by applying a more conservative LOS reduction (1.38 days) with ultrasonic energy compared with conventional techniques informed by Cheng et al. 2016 [13]

[d]deviates from the primary analysis by applying a more conservative daily hospital stay cost (EUR 416) informed by Berto et al. 2012 [33].

**Abbreviations:** ALND: axillary lymph node dissection; BCS: breast-conserving surgery; EUR: Euros; LOS: length of stay.

## Scenario analyses

Cost savings persist within the conservative scenarios evaluating the incremental LOS impact of HARMONIC FOCUS™+ and daily hospital LOS costs. Cost savings are EUR 59,416 and EUR 52,281 for these two scenarios, respectively (Table 3).

# Discussion

## Key findings

This is the first BIA exploring the cost impact of adopting HARMONIC FOCUS™+ in place of monopolar electrocautery for BCS with ALND and mastectomy with ALND, from an Italian hospital perspective. Results demonstrate that adopting HARMONIC FOCUS™+ in place of monopolar electrocautery leads to cost savings that are maintained across sensitivity and scenario analyses, demonstrating the robustness of results, even when conservative input parameters are assumed. Most cost savings are attributable to a reduction in postoperative LOS with the intervention. Furthermore, reduced bed-day requirements have the potential to yield benefits not quantified in our analysis, including reduction in environmental burden [34], as well as increased bed capacity for other admissions, the importance of which was underscored during the COVID-19 pandemic.

To date, four meta-analyses have reported clinical improvements with ultrasonic energy compared with conventional techniques in breast procedures, including lower complication rates, and shorter OR time and postoperative LOS [13, 20, 21, 27]. However, studies included in these meta-analyses spanned country settings and were therefore not appropriate to directly inform a BIA within an Italian setting specifically. These analyses do, however, corroborate the findings of Iovino et al. 2012, the source of clinical inputs to our model [16]. These data are supported by Italian real-world evidence demonstrating improvements in clinical outcomes, for example, reduced seroma formation rate, with HARMONIC FOCUS™+ versus electrocautery following ALND [12]. Another systematic literature review of meta-analyses collated clinical outcome data of ultrasonic energy compared with conventional techniques across various surgical procedures (mastectomy, gastrectomy, colectomy, and oral, head and neck surgery) in multiple countries [14]. Outcomes with ultrasonic energy were consistently improved across surgical specialties [14]. The clinical efficacy of HARMONIC FOCUS™+ is likely to be applicable to other countries given that these studies were conducted across multiple settings [13, 20, 21, 27].

One of the meta-analyses suggested that the reduction in OR time compared with conventional techniques was due to the reduction in the need to alternate surgical devices and the reduction in surgical smoke, which improves visibility of the surgical site [14]. Several of the meta-analyses described above, and Iovino et al. 2012, associated the reduction in seroma frequency and postoperative chest wall drainage with the improved precision of dissection and lower thermal injury, leading to a reduced inflammatory response [14, 16, 20]. Furthermore, unlike electrocautery, use of HARMONIC FOCUS™+ does not generate an electrical current, which has been shown to cause muscle contractions and can lead to muscle impaling injuries that cause bleeding and may elicit an inflammatory response [35]. Reduction in seroma frequency and postoperative chest wall drainage is likely to lead to quicker patient recovery responsible for reducing LOS [14].

Other cost analyses of ultrasonic energy, which corroborate the findings of this BIA, have been conducted, albeit within different country settings, populations and/or measures of in-hospital costs [22–24]. One BIA compared an ultrasonic energy device with electrocautery for an annual caseload of breast procedures (mastectomies [n = 100] and lumpectomies [n = 100]) in a Canadian hospital setting. Overall potential net cost savings of CAD 171,966 for the caseload were achieved with ultrasonic energy when considering costs associated with surgery, LOS and complications [24]. In a meta-analysis of 13 studies comparing ultrasonic energy to conventional techniques across surgical specialties, OR costs (comprising costs for devices, consumables and OR time) were statistically lower, by 8.7%, with ultrasonic energy [22]. Furthermore, in a literature review conducted by Hsiao et al. in 2015, cost savings for ultrasonic energy versus conventional techniques in breast, colon and thyroid surgery were estimated to be USD 85–400 per procedure, from a variety of country settings [23]. These clinical and cost analyses suggest that the cost savings associated with the use of HARMONIC FOCUS™+ observed in this BIA may also be achieved in other countries and surgical specialties.

An Italian health technology assessment (HTA) report published in 2014 reported the clinical benefits of using ultrasonic energy in breast surgery, yet called for further evidence to support implementation of ultrasonic energy in clinical guidelines [36]. The findings of this BIA, in conjunction with the described cost analyses and meta-analyses [13, 14, 20–24, 27], address this need and highlight how the implementation of ultrasonic energy in clinical guidelines would yield benefits from a clinical and hospital budget perspective.

## Study strengths

A conservative approach in the design of the model was used. Although cost savings observed in real-world practice may differ from the BIA results, this approach greatly strengthens the robustness of the conclusion. In particular, some complications (e.g. bleeding and infection), which are resource-intensive and have been shown to result in meaningful cost and/or clinical differences between ultrasonic energy and conventional techniques [13, 15, 17, 18, 22, 29], were not considered in our analysis due to a lack of appropriate Italian unit cost data, despite their potential contribution to additional cost. Limiting the scope of our analysis to in-hospital costs is also conservative in favour of the comparator, given that complications are likely to have a downstream economic impact beyond the surgical admission [13].

## Study limitations

It is also important to acknowledge that there are some limitations of this study. Iovino et al. 2012 reported on a mixed cohort of patients who underwent BCS with ALND or mastectomy with ALND, without stratifying the outcomes by procedure [16]. As such, the model did not

distinguish the clinical differences between BCS with ALND and mastectomy with ALND. Therefore, cost savings may differ in settings with an alternative proportion of procedures. However, as the study recruited consecutive eligible patients in an Italian hospital, it is considered a fair representation of the ratio of procedures expected in Italian clinical practice [16]. Furthermore, the larger proportion of BCS procedures compared with mastectomies in Iovino et al. 2012 is in line with current and future trends [16, 37], given that the most recent ESMO guidelines state BCS (with radiotherapy) is the gold standard for patients with early-stage breast cancer [6]. The use of literature published between 2012 and 2016 to inform cost inputs (in the absence of more recently published data for the Italian setting) may also be considered a limitation of the model. However, all costs were inflated to 2020 EUR values in order to reflect contemporary costs.

## Conclusion

HARMONIC FOCUS™+ achieved robust cost savings when used in place of monopolar electrocautery for BCS with ALND and mastectomy with ALND procedures, from an Italian hospital setting. Improved clinical and resource outcomes were also predicted for the annual caseload of breast procedures with HARMONIC FOCUS™+, comprising fewer patients experiencing seroma, shorter duration of postoperative chest wall drainage for patients, more OR time saved, and hospital bed days freed. These findings address the need for more evidence of ultrasonic energy in breast surgery highlighted in the 2014 Italian HTA report, and support its place as a preferred option for cancer patients requiring breast surgery.

## Supporting information

**S1 Table. Parameters and distributions applied in the sensitivity analyses. Abbreviations:** DSA: deterministic sensitivity analysis; EUR: Euros; LOS: length of stay; OR: operating room; RR: risk ratio; PSA: probabilistic sensitivity analysis.
(DOCX)

## Acknowledgments

The authors acknowledge William Marsh, PhD, from Costello Medical, UK, for support in designing the model with input and direction from authors, and Molly Atkinson, BSc, from Costello Medical, UK, for support in medical writing of the manuscript with input and direction from authors.

## Author Contributions

**Conceptualization:** Alessandra Piemontese, Thibaut Galvain, Lirazel Swindells, Vito Parago, Giovanni Tommaselli, Nadine Jamous.

**Formal analysis:** Alessandra Piemontese, Thibaut Galvain, Lirazel Swindells, Vito Parago, Giovanni Tommaselli, Nadine Jamous.

**Methodology:** Alessandra Piemontese, Thibaut Galvain, Lirazel Swindells, Vito Parago, Giovanni Tommaselli, Nadine Jamous.

**Writing – original draft:** Alessandra Piemontese, Thibaut Galvain, Lirazel Swindells, Giovanni Tommaselli, Nadine Jamous.

**Writing – review & editing:** Alessandra Piemontese, Thibaut Galvain, Lirazel Swindells, Vito Parago, Giovanni Tommaselli, Nadine Jamous.

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
