## [Decision Letter · Decision Letter 0]

10 Feb 2022

PONE-D-21-28885Budget impact analysis of HARMONIC FOCUS™+ Shears for mastectomy and breast-conserving surgery with axillary lymph node dissection compared with monopolar electrocautery from an Italian hospital perspectivePLOS ONE

Dear Dr. Piemontese,

Thank you for submitting your manuscript to PLOS ONE. After careful consideration, we feel that it has merit but does not fully meet PLOS ONE’s publication criteria as it currently stands. Therefore, we invite you to submit a revised version of the manuscript that addresses the points raised during the review process.

We look forward to receiving your revised manuscript.

Kind regards,

Chun Chieh Yeh, M.D., Ph.D.

Academic Editor

PLOS ONE

https://journals.plos.org/plosone/s/fileid=ba62/PLOSOne_formatting_sample_title_authors_affiliations.pdf".

“This study was funded by Johnson & Johnson. Support for third-party writing assistance for this article, provided by Costello Medical, UK, was funded by Johnson & Johnson in accordance with Good Publication Practice (GPP3) guidelines (http://www.ismpp.org/gpp3).”

“I have read the journal's policy and the authors of this manuscript have the following competing interests: TG, NJ, VP, AP and GAT are paid employees of Johnson & Johnson. LS is a paid employee of Costello Medical, UK, who was contracted by Johnson & Johnson to conduct the assist in developing the model and provide medical writing support in preparation of the manuscript.”

Additional Editor Comments:

Thank you for submitting this remarkable work. After comprehensive reviewing, the referees gave some comments to your work. We wish you could provide specific and timely response and revisions for their comments. Decision will be made based on your response. Thanks.

Reviewers' comments:

Reviewer's Responses to Questions

**Comments to the Author**

1. Is the manuscript technically sound, and do the data support the conclusions?

Reviewer #1: Yes

Reviewer #2: Yes

2. Has the statistical analysis been performed appropriately and rigorously? 

Reviewer #1: Yes

Reviewer #2: Yes

3. Have the authors made all data underlying the findings in their manuscript fully available?

Reviewer #1: Yes

Reviewer #2: Yes

4. Is the manuscript presented in an intelligible fashion and written in standard English?

Reviewer #1: Yes

Reviewer #2: Yes

5. Review Comments to the Author

Reviewer #1: A good article with good statistical analysis on a certo common surgical operation. I think the article Is writer in a good english.the article could have a big impact on guide Lines and influenze the choiche of the Harmonic scalpel for axillary dissection.

Reviewer #2: The Authors performed a really interesting study Budget impact analysis of HARMONIC FOCUS™+ Shears for mastectomy and breast-conserving surgery with axillary lymph node dissection compared with monopolar electrocautery. It is in interesting topic. The paper is well written and interesting in all its field demonstrating a remarkable experience in the treatment of breast cancer.

In my opinion in patients undergoing surgery for breast cancer, one of the unsolved problems is the seroma formation. In order to better analyse this topic, I suggest considering the paper:

" Gambardella C, Clarizia G, Patrone R, Offi C, Mauriello C, Romano R, Filardo M, Conzo A, Sanguinetti A, Polistena A, Avenia N, Conzo G. Advanced hemostasis in axillary lymph node dissection for locally advanced breast cancer: new technology devices compared in the prevention of seroma formation. BMC Surg. 2019 Apr 24;18(Suppl 1):125. doi: 10.1186/s12893-018-0454-8."

“Docimo G, Limongelli P, Conzo G, Gili S, Bosco A, Rizzuto A, Amoroso V, Marsico S, Leone N, Esposito A, Vitiello C, Fei L, Parmeggiani D, Docimo L. Axillary lymphadenectomy for breast cancer in elderly patients and fibrin glue. BMC Surg. 2013;13 Suppl 2(Suppl 2):S8. doi: 10.1186/1471-2482-13-S2-S8. Epub 2013 Oct 8. PMID: 24266959; PMCID: PMC3851152.”

“Parisi S, Ruggiero R, Gualtieri G, et al. Combined LOCalizer™ and Intraoperative Ultrasound Localization: First Experience in Localization of Non-palpable Breast Cancer. In Vivo. 2021;35(3):1669-1676. doi:10.21873/invivo.12426”

6. PLOS authors have the option to publish the peer review history of their article (what does this mean?). If published, this will include your full peer review and any attached files.

Reviewer #1: No

Reviewer #2: No

---

## [Author Response · Author response to Decision Letter 0]

4 Mar 2022

Associate Editor:

General comments: Thank you for submitting this remarkable work. After comprehensive reviewing, the referees gave some comments to your work. We wish you could provide specific and timely response and revisions for their comments. Decision will be made based on your response. Thanks.

Authors’ response: Thank you very much for your positive feedback on our manuscript. In line with the helpful comments from peer review, we have revised our manuscript accordingly. Please find these addressed point-by-point below.

Authors’ response: We have thoroughly reviewed the provided guidance and have made minor formatting changes throughout the manuscript, including adjusting headings and abbreviating journal names within the bibliography, to ensure that it meets PLOS ONE’s style requirements. We have also named each of the resubmitted files as per the provided guidance.

2. Thank you for stating the following financial disclosure: “This study was funded by Johnson & Johnson. Support for third-party writing assistance for this article, provided by Costello Medical, UK, was funded by Johnson & Johnson in accordance with Good Publication Practice (GPP3) guidelines (http://www.ismpp.org/gpp3).” Please state what role the funders took in the study. If the funders had no role, please state: ""The funders had no role in study design, data collection and analysis, decision to publish, or preparation of the manuscript."" If this statement is not correct you must amend it as needed. Please include this amended Role of Funder statement in your cover letter; we will change the online submission form on your behalf.

Authors’ response: The Role of Funder statement should read as follows: “This study was funded by Johnson & Johnson. Alessandra Piemontese, Thibaut Galvain, Vito Parago, Giovanni A. Tommaselli and Nadine Jamous are all employees of Johnson & Johnson. Support for third-party writing assistance for this article, provided by Costello Medical, UK, was funded by Johnson & Johnson in accordance with Good Publication Practice (GPP3) guidelines (http://www.ismpp.org/gpp3).”

3. Thank you for stating the following in the Competing Interests section: “I have read the journal's policy and the authors of this manuscript have the following competing interests: TG, NJ, VP, AP and GAT are paid employees of Johnson & Johnson. LS is a paid employee of Costello Medical, UK, who was contracted by Johnson & Johnson to conduct the assist in developing the model and provide medical writing support in preparation of the manuscript.” Please confirm that this does not alter your adherence to all PLOS ONE policies on sharing data and materials, by including the following statement: ""This does not alter our adherence to PLOS ONE policies on sharing data and materials.” (as detailed online in our guide for authors http://journals.plos.org/plosone/s/competing-interests). If there are restrictions on sharing of data and/or materials, please state these. Please note that we cannot proceed with consideration of your article until this information has been declared. Please include your updated Competing Interests statement in your cover letter; we will change the online submission form on your behalf.

Authors’ response: The Competing Interests statement should read as follows: “I have read the journal's policy and the authors of this manuscript have the following competing interests: TG, NJ, VP, AP and GAT are paid employees of Johnson & Johnson. At the time of writing the manuscript, LS was a paid employee of Costello Medical, UK, who was contracted by Johnson & Johnson to conduct the assist in developing the model and provide medical writing support in preparation of the manuscript. This does not alter our adherence to PLOS ONE policies on sharing data and materials.”

Authors’ response: There is no additional, repository information that needs to be made available prior to publication. All model inputs and their sources (including citations), and a detailed description of the model design and structure, have been provided within the manuscript.

5. PLOS requires an ORCID iD for the corresponding author in Editorial Manager on papers submitted after December 6th, 2016. Please ensure that you have an ORCID iD and that it is validated in Editorial Manager.

Authors’ response: The ORCID iD for the corresponding author is: 0000-0002-7397-3546. 

Reviewer 1:

General comments: A good article with good statistical analysis on a certo common surgical operation. I think the article Is writer in a good english.the article could have a big impact on guide Lines and influenze the choiche of the Harmonic scalpel for axillary dissection. 

Authors’ response: We thank you for your thorough review of this manuscript and generous feedback. 

Reviewer 2:

General comments: The Authors performed a really interesting study Budget impact analysis of HARMONIC FOCUS™+ Shears for mastectomy and breast-conserving surgery with axillary lymph node dissection compared with monopolar electrocautery. It is in interesting topic. The paper is well written and interesting in all its field demonstrating a remarkable experience in the treatment of breast cancer.

Authors’ response: We thank you for your thorough review of this manuscript and generous feedback. Responses to your comments can be found below.

1. In my opinion in patients undergoing surgery for breast cancer, one of the unsolved problems is the seroma formation. In order to better analyse this topic, I suggest considering the paper:

a) " Gambardella C, Clarizia G, Patrone R, Offi C, Mauriello C, Romano R, Filardo M, Conzo A, Sanguinetti A, Polistena A, Avenia N, Conzo G. Advanced hemostasis in axillary lymph node dissection for locally advanced breast cancer: new technology devices compared in the prevention of seroma formation. BMC Surg. 2019 Apr 24;18(Suppl 1):125. doi: 10.1186/s12893-018-0454-8."

b) “Docimo G, Limongelli P, Conzo G, Gili S, Bosco A, Rizzuto A, Amoroso V, Marsico S, Leone N, Esposito A, Vitiello C, Fei L, Parmeggiani D, Docimo L. Axillary lymphadenectomy for breast cancer in elderly patients and fibrin glue. BMC Surg. 2013;13 Suppl 2(Suppl 2):S8. doi: 10.1186/1471-2482-13-S2-S8. Epub 2013 Oct 8. PMID: 24266959; PMCID: PMC3851152.”

c) “Parisi S, Ruggiero R, Gualtieri G, et al. Combined LOCalizer™ and Intraoperative Ultrasound Localization: First Experience in Localization of Non-palpable Breast Cancer. In Vivo. 2021;35(3):1669-1676. doi:10.21873/invivo.12426”

Authors’ response: To better recognise the burden of seroma formation on patients undergoing surgery for breast cancer, we have updated the Introduction (page 4, lines 71–74) and Discussion (pages 12–13, lines 256–259) sections to highlight its incidence and potential impact, including delays to adjuvant treatments, and reduced rates observed with HARMONIC FOCUS™+ versus electrocautery, respectively, with reference to Gambardella (2019) and Docimo (2013), as follows:

“…increased risk of complications including lymphedema and seroma,[8, 10] the latter of which affects up to 85% of patients undergoing ALND and is a source of significant morbidity.[11, 12] Seroma formation may also delay adjuvant treatments such as chemotherapy and, as a result, affect oncological outcomes.[12]”

“These data are supported by Italian real-world evidence demonstrating improvements in clinical outcomes, for example, reduced seroma formation rate, with HARMONIC FOCUS™+ versus electrocautery following ALND.[12]”

We have not however incorporated Parisi (2021) into the manuscript as, after thoroughly reviewing the study, we were unable to identify any results pertaining to, or any mention of, seroma formation in patients undergoing surgery for breast cancer. We therefore feel that Parisi (2021) does not bear sufficient relevance to the topic of seroma formation, or the manuscript overall, to warrant its inclusion. However, if you have any suggestions for where Parisi (2021) might fit within the manuscript, please do let us know and we would be more than happy to take them into consideration.

---

## [Decision Letter · Decision Letter 1]

6 May 2022

Budget impact analysis of HARMONIC FOCUS™+ Shears for mastectomy and breast-conserving surgery with axillary lymph node dissection compared with monopolar electrocautery from an Italian hospital perspective

PONE-D-21-28885R1

Dear Dr. Piemontese,

We’re pleased to inform you that your manuscript has been judged scientifically suitable for publication and will be formally accepted for publication once it meets all outstanding technical requirements.

Kind regards,

Chun Chieh Yeh, M.D., Ph.D.

Academic Editor

PLOS ONE

Additional Editor Comments (optional):

Thanks for your timely revision. Based on your revised work, our referee considered your work if worthy of publication at its current content. Congratulation again for your great work.

Reviewers' comments:

Reviewer's Responses to Questions

**Comments to the Author**

1. If the authors have adequately addressed your comments raised in a previous round of review and you feel that this manuscript is now acceptable for publication, you may indicate that here to bypass the “Comments to the Author” section, enter your conflict of interest statement in the “Confidential to Editor” section, and submit your "Accept" recommendation.

Reviewer #2: All comments have been addressed

2. Is the manuscript technically sound, and do the data support the conclusions?

Reviewer #2: Yes

3. Has the statistical analysis been performed appropriately and rigorously? 

Reviewer #2: Yes

4. Have the authors made all data underlying the findings in their manuscript fully available?

Reviewer #2: Yes

5. Is the manuscript presented in an intelligible fashion and written in standard English?

Reviewer #2: Yes

6. Review Comments to the Author

Reviewer #2: (No Response)

7. PLOS authors have the option to publish the peer review history of their article (what does this mean?). If published, this will include your full peer review and any attached files.

Reviewer #2: **Yes: **Claudio Gambardella

---

## [Editor Report · Acceptance letter]

13 Jun 2022

PONE-D-21-28885R1 

Budget impact analysis of HARMONIC FOCUS^™^+ Shears for mastectomy and breast-conserving surgery with axillary lymph node dissection compared with monopolar electrocautery from an Italian hospital perspective 

Dear Dr. Piemontese:

I'm pleased to inform you that your manuscript has been deemed suitable for publication in PLOS ONE. Congratulations! Your manuscript is now with our production department. 

Kind regards, 

on behalf of

Dr. Chun Chieh Yeh 

Academic Editor

PLOS ONE